# A Study of the Influence of Thermoactivated Natural Zeolite on the Hydration of White Cement Mortars

**DOI:** 10.3390/ma17194798

**Published:** 2024-09-29

**Authors:** Ventseslav Stoyanov, Vilma Petkova, Katerina Mihaylova, Maya Shopska

**Affiliations:** 1Department of Technology and Construction Management, Faculty of Construction, University of Structural Engineering and Architecture “Lyuben Karavelov”, 175 Suhodolska Str., 1373 Sofia, Bulgaria; vensy.stoyanov@vsu.bg; 2Department of Safety Management and Prevention, Faculty of Fire Safety and Civil Protection, Academy of Ministry of Interior, 171 Pirotska Str., 1309 Sofia, Bulgaria; 3Institute of Mineralogy and Crystallography “Acad. Ivan Kostov”, Bulgarian Academy of Sciences, Acad. G. Bonchev Str., Block 107, 1113 Sofia, Bulgaria; kate_mih@imc.bas.bg; 4Institute of Catalysis, Bulgarian Academy of Sciences, Acad. G. Bonchev Str., Bldg. 11, 1113 Sofia, Bulgaria; shopska@ic.bas.bg

**Keywords:** cement mortars, zeolite, river sand, cement hydration, pozzolanic activity, CSH and CSAH phases

## Abstract

One trend in the development of building materials is the partial or complete replacement of traditional materials that have a high carbon footprint with eco-friendly ecological raw materials and ingredients. In the present work, the influence of replacing cement with 10 wt% thermally activated natural zeolite on the structural and physical-mechanical characteristics of cured mortars based on white Portland cement and river sand was investigated. The phase compositions were determined by wavelength dispersive X-ray fluorescence (WD-XRF) analysis, X-ray powder diffraction (PXRD), diffuse reflectance infrared Fourier transformed spectroscopy (DRIFTS), and scanning electron microscopy (SEM), as well as thermogravimetric analysis simultaneously with differential scanning calorimetry (TG/DTG-DSC). The results show that the incorporation of zeolite increases the amount of pores accessible with mercury intrusion porosimetry by about 40%, but the measured strengths are also higher by over 13%. When these samples were aged in an aqueous environment from day 28 to day 120, the amount of pores decreased by about 10% and the compressive strength increased by nearly 15%, respectively. The microstructural analysis carried out proves that these results are due to hydration with a low content of crystal water and the realization of pozzolanic reactions that last over time. Replacing some of the white cement with thermally activated natural zeolite results in the formation of a greater variety of crystals, including new crystalline CSH and CSAH phases that allow better intergrowth and interlocking. The results of the investigations allow us to present a plausible reaction mechanism of pozzolanic reactions and of the formation of new crystal hydrate phases. This gives grounds to claim that the replacement of part of the cement with zeolite improves the corrosion resistance of the investigated building solutions against aggressive weathering.

## 1. Introduction

The increased rate of global urbanization in the past decades has led to the widespread implementation of cement-based materials in the construction sector [1]. Cement production, however, is associated with a very high environmental impact due to the amount of carbon dioxide (CO_2_) released in the atmosphere during the manufacturing process (clinker production—limestone calcination), making it an unsustainable material in the long run and contributing greatly to climate change [2,3]. As such, in 2009, the Cement Technology Roadmap identified four levers for reducing emissions in the cement industry [4]: thermal and electric efficiency, use of alternative fuels, carbon capture and storage, and clinker substitution. Substitution of clinker with diverse supplementary cementitious materials (SCMs) is considered to be the most favorable among the identified levers due to its low economic and performance impacts on cement and concrete production [5]. 

The use of natural pozzolans (tuffs, volcanic ash, perlites, pumices, zeolites) as SCMs for modern cement and concrete production and their impact on the hardened product’s properties have long since been investigated by different researchers [3,5,6,7,8,9,10,11,12,13]. Standardized methods are applied to assess their chemical and physical properties, such as loss on ignition, alkali content, fineness, increase in drying shrinkage, soundness, etc. [14]. The main advantage of SCMs substitution is the pozzolanic effect, which is associated with capability of providing amorphous silica that react with Ca(OH)_2_ in the presence of water [10,12,15,16]. The pozzolanic activity includes two parameters, that is, the maximal amount of Ca(OH)_2_ that a pozzolan can combine, and the rate at which such combination occurs [15,16]. Pozzolan substitution results in a small pozzolanic reaction (less hydration products) at the early stage. With an increase in age, the hydration of cementitious materials increases and the porosity of the mortars is reduced [16]. This significantly improves the strength, which is determined with respect to strength development by standard test procedures for evaluating the activity index of pozzolans [14,17].

Compared to volcanic ash, tuff, pumice, and other natural pozzolans, zeolite offers unique benefits as a supplementary cementitious material [7]. Zeolite addition has a multifaceted impact on cement pastes, mortar, and concrete. It lowers the heat generated during hydration, enhances water resistance, improves the balance between bending and compressive strength, increases the need for water and subsequent shrinkage, and raises the apparent viscosity of the mixture. The durability of zeolite containing cement mixtures is improved due to decreased amounts of C_3_A- and CaO-containing hydration products with lower leachability, as well as reduced SO_3_ and diffusion processes such as salt migration and efflorescence [15,16,18].

In the past, natural stone was one of the most commonly used building materials due to its durability and aesthetic appeal. It was used not only for construction purposes but also for creating rich ornaments and decorations on the facades of buildings. One approach to preserving architectural heritage is to create and exhibit replicas of the architectural elements placed on the facades of buildings. The requirements for materials for this use are to have good processability, high durability, weather resistance, and the possibility of being made in colors that will last over time. A potential choice is a mortar based on a hydraulic binder. To increase its resistance, increased amounts of cement, the addition of pozzolanic additive and the preparation of as dense a composition as possible are needed. The first is solved by using white Portland cement, the second by using a white pozzolanic additive of the metakaolin type (white powder obtained by calcination of kaolin clay), diatomaceous earth, rice husk ash, etc., and the third by a chemical additive based on polycarboxylates, which, however, does not change the coloring characteristics.

The potential for making decorative mortars based on blended binder, composed from white Portland cement and clinoptilolite, leads to interest in investigating the influence of zeolite in the process of hardening. The numerous benefits of this technology are due to the specific rheological behavior (high fluidity) of fresh mixes, which are characterized by the addition of large amounts of active and/or inert mineral additives, low water-to-binder ratio, and modified polycarboxylate-based water-reducing admixture [19,20,21].

In our previous works, we have studied the effects of zeolite addition in cement mortars containing high amounts of additives, along with investigations into the structure and thermal behavior of white cement containing river sand and marble powder [22,23,24,25,26].

The goal of the current work is to study the effect of the addition of natural Bulgarian clinoptilolite into the structure of white cement mortars during hydration and curing at 28 and 120 days, without the presence of any other additives. This effect is established by studying (i) the formation and development of the pore space in cement mortars, (ii) the macro- and microstructure, and (iii) the phase formation of new CSH and CSAH crystal hydrates. Samples containing 10% zeolite and different water-to-cement ratios were prepared so as to better study the process of phase formation. 

The investigated parameters and their effects were studied with the following methods: wavelength dispersive X-ray fluorescence (WD-XRF) analysis, powder X-ray diffraction (PXRD), scanning electron microscopy (SEM), diffuse reflectance infrared Fourier transformed spectroscopy (DRIFTS), thermogravimetric analysis simultaneously with differential scanning calorimetry (TG/DTG-DSC), and physical-mechanical methods for obtaining the following properties: density after immersion, adsorption after immersion, compressive strength, and porosity.

## 2. Materials

The objects of research in the present work are cement mortars with different ratios of the solid components. In this research, white Portland cement CEM I 52.5 N was used, produced by Devnya Cement (Devnya, Bulgaria). The mineral composition, calculated by the Bogue method, was as follows (in wt%): C_3_S—72.13; C_2_S—15.28; C_3_A—5.23; C_4_AF—0.61. The river sand (clean, washed, and dried) that was used for the preparation of the mortars has a fineness modulus FM = 2.7 (EN 12620) and shape index—4.6% (EN 933-4), i.e., spheroid particles, over 85.0% SiO_2_ content [27]. 

The zeolite addition was milled, (0–0.8 mm) natural clinoptilolite tuff, mined from Beli Plast deposit, Bulgaria. It contains about 80 wt% clinoptilolite and admixtures of montmorillonite, biotite, celadonite, low-cristobalite, calcite, quartz, and feldspars. The amount of clinoptilolite in the composition of the solutions is 10%. The limitation of 10% replacement of cement with clinoptilolite was necessary to avoid the dominance of the zeolite color and to avoid significant changes in the rheological behavior of the systems. 

The zeolite (natural clinoptilolite) was thermally activated at 250 °C [7,23]. Thermally activated zeolites are classified in 3 categories [28]: (i) they preserve their skeletal structure but the zeolite water is dehydrated, leading to the opening of interlayer spaces and channels in the zeolite. Eventually, they can reabsorb water on cooling and restore their original state; (ii) substantial distortion in the skeletal structure while preserving the ability to rehydrate; (iii) breakdown of the T-O-T chains (where T can be a Si or Al atom) during the dehydration process while achieving a significant distortion in the topology of the zeolite skeletal structure and loss of the ability to rehydrate. It is assumed in the current study that most of the water will enter the skeletal structure upon hydration of the cementitious minerals following the thermal activation of zeolite up to 250 °C. This water will then be used for the formation of CSH/CSAH phases and ongoing hydration over time, thereby promoting the pozzolanic activity of zeolite. The present work aims to find evidence supporting this assumption in the findings of the analytical techniques applied.

To support the choice of materials, a short economic study was made on the current prices of some cement substitutes in the commercial network in Bulgaria. The data are presented in Table 1. 

The data in Table illustrate the feasibility of using an accessible, inexpensive, natural material with a high major component content such as natural clinoptilolite.

The experiments were carried out with two cement mortars (Table 2) following the mass ratio of 1:3:0.5 used in EN 196-1 to determine the compressive strength and, optionally, the flexural strength using prismatic samples with sizes 40 × 40 × 160 mm. The zeolite addition was always introduced into the mixtures together with distilled water. Immediately after the mixing procedure (according to EN 196-1 at 20 °C and 65% RH), the workability of the fresh mortar was evaluated using the standardized measurement of consistence [29]. The spreading diameter for mortars was measured by using a truncated cone, based on a flow table test (EN 1015-3) [30]. The results indicate that the addition of clinoptilolite-based zeolite reduces the spreading diameter from 111 ± 2 mm for Ms to 107 ± 3 mm Mz.

After casting, the samples (6 prisms, 40 × 40 × 160 mm) were stored in the molds for 1 day in a moist atmosphere (>95% RH and 20 °C). Once demolded, the samples were kept under water (20 °C) until strength testing (28 and 120 days). The compressive strengths at 28 and 120 days of water curing were measured according to EN 196-1 [29].

## 3. Methods

Broken parts of samples with mass of 2.0 ± 0.3 mg were used to measure the porosity by the mercury intrusion porosimetry method (MIP) using Carlo Erba Porosimeter Mod. 1520 (Carlo Erba Strumentazione, Rodano (Milan), Italy), with the pressure range 0.10–150 MPa, corresponding to the pore size range 50–15,000 nm.

Wavelength dispersive X-ray fluorescence (WD-XRF) analysis was performed using spectrometer WD-XRF Super-mini 200 (Rigaku Corporation, Tokyo, Japan) (operating at 50 kV and 4 mA, 200 W X-ray tube with Pd-anode, 30 mm^2^) under a helium atmosphere. Two different X-ray detectors were made use of: a gas flow proportional counter for light elements and a scintillation counter for heavy elements. Three analyzing crystals were utilized (according to the wavelength range): LIF 200 (for Ti-U), PET (for Al-Ti), and RX25 (for F-Mg). The samples were prepared as tablets with CERE-OX-BM-0002-1 powder. Rigaku’s built-in software package “ZSX” ver. 7.67 was employed for the processing of the data.

The PXRD patterns were performed on an X-ray powder diffractometer D2 Phaser (Bruker AXS GmbH, Karlsruhe, Germany) using CuKα radiation (λ = 0.15418 nm) (operating at 30 kV, 10 mA) from 5 to 80° 2θ with a step of 0.05° (grinded sample with weight—1.0 ± 0.1 mg and particle sizes below 0.075 mm). Phase identification was carried out using various search-match software programs, as well as by using data from the Inorganic Crystal Structure Database (ICSD) [31].

Experiments were conducted to characterize the macro-, meso-, and microstructure of the investigated samples. The macrostructure was captured using a Nikon D3300 SLR digital camera with a Nikon AF-P 18–55 mm VR lens (Nikon Corporation, Tokyo, Japan), effective pixels—24.2 MP, processor—EXPEED 4, capture sensor—CMOS 23.5 × 15.6 mm, light sensitivity (ISO 100)—12,800 in 1 EV steps. The microstructure was observed with Philips PH, Model 515, regime of secondary electron emission. Following a 12 h drying process at 60 ± 5 °C, the fracture fragments of the samples, which had a roughly flat surface measuring 10 × 10 mm, were covered with a thin layer of gold.

Additionally, DRIFTS was applied for the analysis of the samples. It was used in order to increase the sensitivity of the IR analysis, which, due to the measurement method, required the use of 100% of the studied substance without any dilution. The samples were studied with a spectrometer Nicolet 6700 (Thermo Electron Corporation, Waltham, MA, USA). A diffuse-reflectance accessory Collector II (Thermo Electron Corporation, Waltham, MA, USA) was used. The spectra were recorded at a resolution of 4 (data spacing 1.928 cm^−1^) and 100 scans.

Thermal studies (TG/DTG-DSC) were performed with a Setsys Evolution 2400 thermo-analytical complex (THEMYS), SETARAM, Caluire-et-Cuire, France. A dynamic heating regime was selected in the temperature range from room temperature (RT) to 1300 °C with a heating rate of 10 °C min^−1^. The mass of the samples for thermal characterization analysis was 18.0 ± 2.0 mg (mass resolution of 0.02 µg). The calorimetric measurements were performed with a B-type sensor with a temperature measurement range up to 1600 °C and a DSC rod with a resolution of 1.0 μW. The experiments were performed in a gas atmosphere of static air. Stabilized ceramic crucibles with a volume of 100 µL were used, respecting the requirement of filling with a low reaction layer to eliminate diffusion difficulties.

## 4. Results 

### 4.1. Wavelength Dispersive X-ray Fluorescence Analysis

Table 3 presents the results of the semi-quantitative XRF measurements of the major component contents of the raw materials and the cement mortars.

The results of the XRF analysis of the samples Ms28, Ms120, Mz28, and Mz120 present some trends in the major component oxide contents—CaO, SiO_2_, Al_2_O_3_, Na_2_O, K_2_O, SO_3_, etc. The data in Table 2 are the result of the cement mortars obtained by mixing white cement, river sand, and natural zeolite. The highest contents were SiO_2_, 50.73–54.23%; CaO, 27.68–30.84%; and Al_2_O_3_, 9.27–9.64%. The deviation from the average values for each of the major components is within approximately ±2%, which is evidence of compositional consistency. There is a greater increase in the amount of Al_2_O_3_, from 3.45% in the white cement to 9.46% average in the cement mortars. This result is related to the higher content of this component in river sand [32,33]. It is assumed that Al_2_O_3_ will make a significant contribution to the phase formation of CSAH hydrates in the hydration process.

The semi-quantitative results of the XRF analysis prove the presence of major oxides in the starting components used for the preparation of the mortars. The main prerequisites for their phase formation are expected to be the proportions between them, the influence of the water-to-cement ratio and the presence of zeolite, which will have an impact on the hydration process.

### 4.2. Physical-Mechanical Properties

The hydration process of cementitious materials is essential for the stability and strength of the resulting mortars. In their production, not only is the type of basic materials used essential but so are additives, inert fillers, and waste, which reduce the cost of their production but without compromising the quality of the final products. Therefore, the first analyses that provide information on the effectiveness of the technologies for obtaining building mortars are physical-mechanical studies.

The following physical-mechanical properties were measured: density after immersion, water adsorption after immersion, bending strength compressive strength, and pore volume at 28 days and 120 days. The measuring method and the average values of the properties are shown in Table 3. Due to the samples’ varying density, the parameter “adsorption after immersion” is calculated in volume percentages.

From the data in Table 4, the high density of the Ms and Mz systems is reported. For all compositions, with increasing age, the densities remain almost unchanged, with changes within the margin of error. The replacement of 10% cement with zeolite reduces the parameter “density after immersion” due to air-entraining (increase in both the number and sizes of air bubbles). The addition of the zeolite leads to a reduction of the parameter “bending strength”, which may be the result of air-entraining or the presence of unreacted soft zeolitic particles. 

In order to determine the “absorption after immersion” (Table 4), firstly, the samples are dried in an oven at 50 ± 5 °C for 3 days and their weight is measured. Then, they are immersed in water for 5 days, which ensures full water absorption (increase in mass was less than 0.5% of the heavier mass), and their weight is measured again. The amount of water permeated in the Mz system was greater by almost 150% compared to that of Ms. The reason for these results is the addition of clinoptilolite. The water absorption, measured at 28 days of curing, of both Ms and Mz samples is relatively large for this type of material, suggesting possibilities for obtaining a denser structure through a precise selection of fractions of inert/active fillers [34].

The formed structure of Mz samples is dense, which is determined when comparing the pure volume of the samples, recalculated from the measured value of the pore volume and density after immersion at 28 days of curing—21.1 mm^3^/cm^3^ (sample Ms); 30.0 mm^3^/cm^3^ (sample Mz) (Table 4). However, these samples have an open porosity, which allows for the penetration of water, causing a slow, continuous hydration of the cement grains. The high adsorption of the samples with natural zeolite and river sand can be explained by the high water absorption capacity of zeolite particles [35].

According to Table 4, the standard cement mortar Ms’s 28-day compressive strength is similar to the factory’s stated value. This suggests that the sand utilized is comparable to EN 196-1, the CEN Standard sand. At 120 days, the pore volume drops by 6.8% and the newly formed hydrates fill the porous area, increasing the compressive strength by 7.6%.

Similarly, for samples Mz28 and Mz120, a 15.1% increase in the compressive strength was observed. The values of the measured parameters indicate the formation of a thin structure of hardened samples Mz, which are characterized by a 0.45 water-to-cement ratio.

The compressive strength of both samples Ms28 and Mz28 is very high and continues to grow over time. The use of zeolite as a partial binder substitute increases the 28-day (Mz28) compressive strength in comparison with the same parameter for the Ms28 sample, with approximately 13.0% (Figure 1a). At 120 days of curing (samples Ms120 and Mz120), the difference between the compressive strengths is increased by 20.8%.

The results regarding the changes in pore volume have a different pattern compared to the “compressive strength” parameter. The pore volume in the series (Ms28–Ms120 and Mz28–Mz120) decreased by 6.8% for the Ms samples and similarly by 9.2% for the Mz samples (Figure 1b). This result is logical considering that in the course of hydration and deferred hydration, the pores are gradually filled with the crystallites of the CSH/CSAH phases. The difference in the pore volume results between the two series is also significant, with the Ms28–Mz28 pair showing a 43.8% increase in pore volume and the Ms120–Mz120 pair showing a 40.0% increase. With the addition of 10% thermoactivated clinoptilolite, it can be assumed that the observed changes are due to the porous structure of the zeolite [36]. 

On one hand, the formation of a dense and robust structure is observed, but on the other hand, an increase in the pore volume is also noted, in which the role of clinoptilolite is considered as an opportunity to increase the pore space and growth potential of the CSH/CSAH phases.

The reported effects of the influence of natural clinoptilolite on the microstructure of the cement mortars can be confirmed by microscopic analysis—SEM.

### 4.3. Macro- and Microstructure of Mortars—Scanning Electron Microscopy (SEM)

The results of the macro- and microstructural analysis of the Ms and Mz samples are presented in Figure 2, Figure 3, Figure 4 and Figure 5.

The macrostructure of the Ms-series samples with sand is characterized by single-grained particles visible to observation without magnification. Their shape is variable and irregular, and most are of different colors with shades between gray, dark gray, and yellow. The structure is well homogenized; however, characteristically, the air pores are easily visible, entrained during sample preparation and leaving empty circular cavities. It can be determined that the macrostructure is robust and dense, despite the multiple air pores. The sample does not crumble or spatter on contact.

Even at 28 days of curing, the micrographs in Figure 2 show stable and dense structures; as a result, there are no gaps in the structure that could allow new minerals to form. A fine-grained crystal aggregate with structural characteristics most likely below the limit of PXRD detection is formed by the CSH/CSAH gel. The micrographs also show regions where self-desiccation drying cracks are visible. On the surface of sample Ms28 are easily distinguishable areas of dense surface, probably unaffected by the hydration process. The observed effects are explained by water deficiency as a result of the type and ratio of the mortar components. Areas adjacent to these sites are also reported to be covered by relatively small flat formations and those that resemble the shape of a small rose flower. The main morphological characteristic of the sample is that two-dimensional flat crystals on the surface of the sample are predominant while other CSH/CSAH gel-specific formations are in limited areas and are small in size compared to the adjacent crystal formations. The plate crystals are characteristic of portlandite crystals, while those of the second type are characteristic of small crystal forms of the CSH phases. Similar results have been presented by other authors for building mortars with conventional compositions [13,37].

The micrographs of sample Ms120 clearly show long and fine crystallites covering the entire surface of the sample. This type of crystalline form is very characteristic of ettringite [38,39].

Although the surface of the sample is covered with hydration products, cracking is still visible in the foundation (Figure 3).

Macro images of the surface of the Mz series of samples show the same characteristic grain structure, with well-dispersed sand particles, which give the characteristic mélange structure of the two samples Mz28 and Mz120 (Figure 4 and Figure 5). Sample Mz28 (Figure 4) exhibits larger-sized and more densely spaced air pores, which is consistent with the results of the pore volume measurement, which increases significantly (Figure 1). In sample Mz120 (Figure 5), the size of the air pores visibly decreases while the other characteristics are preserved. The macrostructure is again dense, strong, and crack-free. This result is also correlated with the pore volume measurement results, which are higher compared to the pore volume for the whole Ms series but lower compared to sample Mz28.

The microstructures of Mz28 and Mz120 differ from those of Ms28 and Ms120, with higher surface coverage of Mz samples with hydration products. No unhydrated zones are visible, and the crystallites are in larger volume and in greater variety.

Both two-dimensional plate-shaped crystallites and the characteristic needle-shaped crystals, the formation of druses, as well as a surface covered with a thin veil of honeycomb-like crystalline formations characteristic of tobermorite can be identified [40]. Cavities on the surface of the samples can be distinguished, but not micro-cracks, which is consistent with the results of the physical-mechanical measurements for compressive strength. The increase in the values of this parameter for Mz28 and Mz120 corresponds to the greater variety of crystallites formed during the hydration of the cement mortars. From the micrographs of the zeolite series samples, the addition of the crystallites into the basic framework of the cement mortar is evident [36]. In addition, coalescence of the flat crystallites into adjacent formations is noted. This implies that the compactness of the microstructure is improved, which is probably the result of the implementation of pozzolanic reactions, in the course of which additional calcium-silica crystal hydrates (CSH gel) are formed. Pozzolanic reactions can be carried out with fragments of the zeolite structure (based on amorphous SiO_2_ and Al_2_O_3_) [10], in which secondary hydration between the portlandite and zeolite fragments is possible with the formation of CSH/CSAH phases, which is confirmed by the physical-mechanical and microscopic results.

In conclusion, due to the different cement-to-water ratio and empty space, these areas are filled with crystal hydrates of different morphology: plate crystals (portlandite), needle crystals (ettringite), and fine crystal aggregates (CSH/CSAH gels). They cover the surface of the samples with a fine veil of crystallites, the two-dimensional portlandite tiles being considerably less numerous. It is the relation of the portlandite crystallites that is the main difference in the morphology of the samples. In sample Ms28, they are most massively distributed and cover its surface, which is also represented in the SEM image (Figure 2). In the remaining samples (Figure 3, Figure 4 and Figure 5), the flat portlandite formations are significantly fewer and are embedded in the CSH/CSAH network of crystals. These results can most likely be explained by the influence of clinoptilolite (Mz series), which absorbs water faster than cement, and for this reason, there is not enough water for it at the beginning of hydration. The absorbed moisture from the zeolite then participates in the formation of the new CSH/CSAH-type crystal hydrates as secondary hydration continues in the pores of the samples, compensating for the effects in the earlier stage of the hydration process.

In order to confirm the results of the morphological and physical-mechanical studies, X-ray powder diffraction studies were carried out on the phase composition of the cement mortars of the two series.

### 4.4. X-ray Phase Analysis

The PXRD analysis (Table 5 and Figure 6a–d) shows the presence of two groups of minerals in the studied samples: (i) relict minerals from the initial composition: belite, albite, anorthite, Mg-rich calcite, quartz, and dolomite; and (ii) newly formed minerals: portlandite, yugawaralite, scolecite, ettringite, tobermorite 11A, and hibschite. The results of the complete analysis of the phase composition of the samples are presented in Table 4 and Figure 6a–d. The new phases are from CSH and CSAH gels. The formation of the CSAH phases is possible because of the presence of the cementitious minerals from the starting raw materials and the use of quartz sand as a filler, in the composition of which SiO_2_ and Al_2_O_3_ are present (Table 2). Furthermore, its porous structure creates a basic framework that facilitates the formation of crystal hydrate phases (ettringite, tobermorite 11A, and scolecite) in the cavities and interlayer spaces into which water molecules can be inserted. Subsequently, they are used to carry out secondary hydration processes and grow new crystal hydrates. Two of the phases, yugawaralite and hibschite, are identified only in the Mz series samples. Most CSH are formed in both series because they contain quartz and aluminates imported from cement minerals and river sand.

Inadequate water in the systems causes an incomplete hydration of the raw minerals, which is linked to the identification of phases from the raw materials in the compositions of the samples.

The results of the powder PXRD analysis are in full correlation with the results obtained from the physical-mechanical analyses and the macro- and microstructure of the samples from both series. The hydration conditions and the composition of the materials are a prerequisite for the formation of CSH and CSAH phases. The introduction of clinoptilolite (part of zeolite) is involved in the distribution of the amount of water, and it can be assumed that, due to secondary hydration processes, water is in short supply. The PXRD results identify multiple new phases, but the intensity of the reflections is not well pronounced and it can be assumed that these phases have low crystallinity, consistent with the morphological images. Since the amount of water was predicted at the initial stage of sample preparation, it can be argued that the results obtained are expected and their purpose is to assess the influence of the amount of natural zeolite, and in the subsequent stages of the studies, the amount of zeolite, as well as the water–cement ratio, will be refined.

### 4.5. Diffuse Reflectance Infrared Fourier Transformed Spectroscopy (DRIFTS)

IR spectroscopy is a powerful method for measuring organic and inorganic substances in the liquid or solid aggregate state. This method can be used to study cement composites, allowing the identification of solids in minimum quantities and with fine dispersion [23]. In the present work, measurements were performed with diffuse reflectance infrared Fourier transformed spectroscopy (DRIFTS). The method is suitable for the study of substances with an amorphous structure where other structural methods are inapplicable, such as powder X-ray phase analysis. An advantage of DRIFTS is the possibility to directly measure spectra from powder samples with rough surfaces without prior preparation or as dispersions in IR transparent matrices such as KBr and KCl [41,42]. The collecting optics in the DRIFTS attachment is designed to gather as much of the diffuse reflected light as possible. DRIFTS is a suitable technique for the study of chemical bonds in a molecule, especially for determining the involvement of water molecules in the structure of substances. This makes the method particularly applicable for the characterization of cement mortars, especially for the distribution of structurally bound water in the composition of CSH/CSAH phases as a result of hydration of cement minerals involving natural clinoptilolite [43]. 

The results of the spectroscopic measurements using the DRIFTS technique are presented in Table 6 and Figure 7.

The application of the diffuse-reflectance methodology to the measurement of vibrations in molecules makes it possible to identify the vibrations in molecules and the bonds of individual atoms to the various functional groups and especially those to water. This characterization is important for the investigated cement mortars, since the amount of water used in the solutions is not only important in terms of the economic performance of the process but also relevant for the new phases formed, their diversity, and structural characteristics. In summarizing the results, groups of spectral bands characteristic of silicate, carbonate, sulfate, compounds with hydroxyl ions, and those containing crystallization water were identified. The main group of compounds corresponds to the spectra of the CSH and CSAH gel phases and accordingly combines the vibrations of the tetrahedral SiO_4_ group, the pyroxene SiO_3_ group, and the octahedral AlO_6_ in combination with the water molecules. It is a characteristic feature of the studied system that phases containing Al-O-Si bonds are found both in the starting raw materials, e.g., natural clinoptilolite [22], and in the hydration products of cementitious minerals.

In confirmation of the XRD results, the combination of bands of the tetrahedral SiO_4_^4−^ group (Si-O bond with bands at ν_4_ 566–576 cm^−1^ and ν_3_ 1190 cm^−1^) is assigned to the newly formed CASH gel phases such as hibschite.

The pyroxene group SiO_3_^2−^ (Si-O bond), with the most typical bands at ν_2_ 468–472 cm^−1^ and ν_3_ 1040–1053 cm^−1^, is presented in CSH gel phases—hillebrandite, tobermorite-11A, and scolecite and in one new phase—yugawaralite.

Hibschite and yugawaralite are from the group of CASH gels and are formed only in the cement solution with the addition of natural clinoptilolite. They are characterized by Al-O-Si bonding, which is expressed by the bands at 1029–1039 cm^−1^ (ν_3_ Si-O stretching vibration) and 522–528 cm^−1^ (ν_4_ Al-O-Si bending vibration) for wairakite and yugawaralite [45].

The octahedral AlO_6_^9−^ groups (Al-O bond) from the raw tricalcium aluminate are distributed between the newly-formed monosulphoaluminate and ettringite. Their bands are situated at the spectral ranges 410–430 cm^−1^ and 800–950 cm^−1^. The band at ν_3_ 856 cm^−1^ is typical for ettringite (Figure 7) [45]. The vibrational bands of the AlO_6_^9−^ groups combine with the bands of the sulphate group at ν_4_ 690–800 cm^−1^, ν_3_ 1155 cm^−1^, and ν_1_ 958–991 cm^−1^ and with that of the hydroxyl ion—ν_3_ 3590–3592 cm^−1^.

A second group of bands includes the vibrations of the carbonate ion at ν_2_ 869 cm^−1^ and ν_3_ 1409–1554 cm^−1^, which characterize the spectrum of calcite, a mineral that enters into the composition of white cement.

Using the DRIFTS method, the vibrational fluctuations of the Me-OH bonds are better expressed in the spectra. This is evidenced by the well-formed bands in the ν_3_ region 3600–3750 cm^−1^. In this particular system, this bond is realized both between Ca-OH at ν_3_ 3629–3631 cm^−1^ and between Al/Si-OH in the crystal hydrates of CSH and CSAH phases at ν_3_ 3737 cm^−1^.

The composition of all crystal hydrate phases, formed in the process of hydration of the cement minerals, includes crystallization water. The vibrational bands of water are well expressed in the DRIFTS spectra and represent a broad and well-defined band with multiple insertion bands in the range 3411–3432 cm^−1^. In the ν_2_ region 1616–1666 cm^−1^, a bending vibrational mode of crystal water is identified.

In conclusion, it can be summarized that the microstructure and properties of the cement mortars of the present study containing thermoactivated zeolite as an additive and cement replacement were characterized using a complex of analytical and physical methods. The results obtained allow us to evaluate the contribution of the additive to the modified properties of the solutions in the hydration process.

The DRIFTS spectra identify the vibrational modes of the functional groups that enter into the composition of the cement solution minerals, namely SiO_3_^2−^, SiO_4_^4−^, AlO_6_^9−^, SO_4_^2−^, CO_3_^2−^, and OH^−^, as well as the cross vibrations between Al and Si ions across the oxygen bridges—Al-O-Si. The quantitative characteristics of the obtained spectra are presented in Table 6.

The role of clinoptilolite (part of zeolite) in the present study is explained by the following: (i) influence on the formation of pores in the structure, in which further secondary hydration can develop in the growth of crystal hydrate phases; (ii) participation in pozzolanic reactions, in which excess portlandite is absorbed and new CSH/CSAH phases are formed; (iii) preservation of the physical-mechanical performance of the building solutions, which provides a perspective for further development of the topic for practical applications.

### 4.6. Thermal Analysis TG/DTG-DSC

Thermogravimetric analysis, combined with derivative thermogravimetry and differential scanning calorimetry, is a powerful tool to complement the results of the analytical methods used. Highly sensitive modern instrumentation allows precise measurements of mass loss, heat effects, phase transitions, solid-phase synthesis, and other important characteristics of the samples, which reveal new data and provide new evidence for both the structure of the samples and their thermal behavior in the programmed heating regime. This also defines thermal analysis methods as very suitable both for the identification of structure-phase transformations and as an acid-free method for solid-phase synthesis [51,52]. In the present work, the simultaneous TG/DTG-DSC measurements were performed in an air-gas environment up to 1300 °C and using dynamic heating mode. Cement mortars are systems of inorganic composites mainly based on the oxides of calcium, silica, and aluminum. In these solutions, the chemical elements involved are usually in their higher oxidation state and are therefore not sensitive to oxygen from the air. This allows the experiments to be carried out at high temperatures up to 1200–1300 °C in a static air environment without the risk of additional oxidation reactions or syntheses between the main oxides as the results obtained are in response to the thermal decomposition of the constituent phases. This provides a rationale for using the complex of thermal methodologies to identify and, in this case, to complement the results of the other analytical methods and especially to characterize the process and products of hydration of the studied white cement solutions.

The results of the thermal measurements are presented in Table 7 and in Figure 8a–d. For convenience, Table 7 includes our own data on the thermal decomposition of white Portland cement (applied as a control sample to monitor changes in the solution), which was used to prepare the mortars [23,26]. The analysis of the results is presented in Figure 9a–d with complementary plots of the dynamics of the resulting mass losses in characteristic temperature intervals with thermal reactions.

The analysis of the results of the thermal experiments (Figure 8a–d) and the data from Table 7 allow us to claim that in the temperature interval RT–1300 °C, the recorded thermal reactions can be separated into four stages. In each of them, two, three (500–730 °C), or more (30–200 °C) mutually overlapping thermal reactions can be registered. The determination of the initial and final temperature and mass loss for each of them is performed using the derivative thermogravimetry (DTG curve) data, with the temperatures at their inflection points presented in Figure 8a–d and Table 7.

Based on the results of our previous studies [23,26], the thermal effects in the present study are also shown to be related to the following:Temperature range—30–200 °C—Dehydration of physic sorption water and crystal water from crystal hydrates formed during the hydration stage. Their phase composition was identified by PXRD (Figure 6a–d, Table 5) and SEM (Figure 2, Figure 3, Figure 4 and Figure 5) analyses and includes ettringite, tobermorite, yugawaralite, scolecite, and monosulphoaluminate. The diversity in the crystal hydrate phases explains the overlapping dehydration reactions. It can be noted that mass losses are low and do not exceed 2.50%, with the highest being observed in Mz120. Another characteristic feature is the gradual increase in dehydration temperatures due to densification of the structure, as evidenced by MIP (Figure 1b), SEM (Figure 2, Figure 3, Figure 4 and Figure 5), and physical-mechanical measurements (Figure 1a, Table 4);Temperature range—422–500 °C—It is characterized by a pronounced single thermal reaction occurring independently in the second temperature interval. It is also accompanied by low mass losses of 0.63% to 0.75% and a low-intensity endothermic effect. These mass losses are related to the dehydroxylation of Ca(OH)_2_. The losses are low due to the low water–cement ratio and the additional influence of the thermoactivated zeolite, which absorbs some of the water. The portlandite in the samples is formed upon hydration of the unabsorbed portion of water to form CSH and CSAH crystal hydrates, as evidenced by PXRD (Figure 6a–d), SEM (Figure 2, Figure 3, Figure 4 and Figure 5), and DRIFTS (Figure 7) results. There is a tendency to decrease the mass loss from dehydroxylation of Ca(OH)_2_ in the compositions with zeolite, since its role is precisely to participate in pozzolanic reactions to eliminate hydroxyl ions.Temperature range—500–730 °C—Dehydroxylation of CSH/CSAH, partial decarbonation. This temperature range is most closely related to the hydration of cement solutions and the influence of thermoactivated zeolite. A fraction of the water molecules bind to the calcium-silica/aluminum phases as crystallization water, while another fraction enters the compositions in the form of structurally bound water as OH ions. They form stronger bonds in the crystal lattice of the CSH/CSAH phases. Logically, the stronger bonds are broken at higher temperatures, which fall precisely in the third interval up to 730 °C. The thermal reactions are mutually overlapping, with mass losses ranging from 4.29% to 6.16%, and the effects are endothermic. Based on the PXRD results obtained (Figure 6a–d), it can be assumed that dehydroxylation of ettringite, hillebrandite, hibschite, and tobermorite takes place. The different forms of water molecules as crystallization or structurally bound OH ions were defined in the DRIFTS spectroscopy analyses (Figure 7, Table 6). The thermal experiments confirm these results by complementing the information with mass loss data and thermal decomposition temperatures.Temperature range—640–800 °C—Decarbonation of CaCO_3_. In the high-temperature region, thermal decomposition of a small amount of carbonate ions, mainly from calcite, is observed, as shown by PXRD (Figure 6a–d), SEM (Figure 2, Figure 3, Figure 4 and Figure 5), and DRIFTS (Figure 7, Table 6). Mass losses range close to 3.00%, with a well-defined endothermic effect. It can be noted that the decomposition temperatures of carbonates are shifted towards the higher ones due to diffusion difficulties resulting from the formation of a more solid and dense structure.

In our previous studies [23,26], it was shown that along with the dehydroxylation of OH-ions, a partial decarbonation of thermally more unstable carbonate-containing phases takes place, whose thermal decomposition occurs at lower than usual temperatures, falling in the third interval.

For a clearer illustration of the effect of using thermally activated zeolite in building mortars, the results from the thermal studies are presented in the form of trend plots for the mass losses of the individual processes of dehydration of crystallization water (Figure 9a), dehydroxylation of structurally bound water by Ca(OH)_2_ (Figure 9b), dehydroxylation and partial decarbonation of CSH/CSAH (Figure 9c), and total mass losses (Figure 9d). The data for the Ms series are in orange, for the Mz series in green, and for the white cement used for comparison in blue. The red and blue lines represent the increasing and decreasing trend of ML, respectively, for a given process. The mass loss data dynamics from crystallization water dehydration (Figure 9a) present a decreasing trend for the Ms series and an increasing trend for Mz, particularly well pronounced when comparing the ML data between Mz28—1.24% and Mz120—2.50%, which is the highest value. This means that water is retained when the thermally activated zeolite is used, and it is distributed as crystallization and structurally bound. The ML trends of the structurally bound water (Figure 9c) best illustrate the impact of using zeolite. Both series recorded relatively high MLs between 4.29% to 6.16%. The distinctly higher values are for the Mz series. This is the clearest evidence for the role of zeolite as a natural pozzolan and the possibilities for a secondary continuous hydration. Accordingly, this trend also applies to the total mass losses (Figure 9d), which increase by approximately 5% in the Mz series due to the increase in ML in the third stage, while they remain approximately the same 10–11% for the Ms series. The ML change trend from dehydroxylation of structurally bound water to Ca(OH)_2_ is the least pronounced, but nevertheless, a decreasing tendency can be noted in the Mz series.

## 5. Discussion

According to the results from the complex of analytical methods—MIP, SEM, PXRD, and DRIFTS—the following reaction mechanism scheme of the samples’ hydration have been defined:2Ca_3_SiO_5_ (C_3_S) + 7H_2_O → Ca_3_Si_2_O_7_·4H_2_O (C-S-H gel) + 3Ca(OH)_2_ (fast)(1)
2Ca_2_SiO_4_ (C_2_S) + 5H_2_O → Ca_3_Si_2_O_7_·4H_2_O (C-S-H gel) + Ca(OH)_2_ (slow)(2)

Formation of ettringite:Ca_3_Al_2_O_6_ (C_3_A) + 3CaSO_4_·2H_2_O + 26H_2_O → Ca_6_Al_2_(SO_4_)_3_(OH)_12_·26H_2_O(3)

Formation of hillebrandite (CSH):Ca_2_SiO_4_ (C_2_S) + H_2_O → Ca_2_SiO_3_(OH)_2_(4)

Formation of tobermorite 11A (CSH)—only in compositions with sand or zeolite:2Ca_3_SiO_5_ (C_3_S) + 4SiO_2_ + 6H_2_O → Ca_5_Si_6_O_16_(OH)_2_·4H_2_O (C-S-H gel) + Ca(OH)_2_(5)
3Ca_2_SiO_4_ (C_2_S) + 3SiO_2_ + 6H_2_O → Ca_5_Si_6_O_16_(OH)_2_·4H_2_O (C-S-H gel) + Ca(OH)_2_(6)

Formation of hibschite (CSAH)—only in compositions with sand or zeolite
Ca_3_Al_2_O_6_ + 2H_2_O^−^ + 2SiO_2_ → Ca_3_Al_2_(SiO_4_)_3−x_(OH)_4x_, where x = 1(7)

Formation of scolecite (CSAH):Ca_3_Al_2_O_6_ (C_3_A) + 3SiO_2_ + 5H_2_O → CaAl_2_Si_3_O_10_·3H_2_O + 2Ca(OH)_2_(8)

Formation of yugawaralite (CSAH):Ca_3_Al_2_O_6_ (C_3_A) + 6SiO_2_ + 6H_2_O → CaAl_2_Si_6_O_16_·4H_2_O + 2Ca(OH)_2_(9)

A wider range of newly-formed hydrated phases is recorded during the phase composition identification in the current studies of mortars containing zeolite (clinoptilolite), in contrast with our results from earlier studies of the PXRD analysis of cement mortars without zeolite addition [24,26]. In addition to hillebrandite and scolecite, phases such tobermorite-11A, yugawaralite, and hibschite are formed in the presence of up to 10% clinoptilolite in the cement, even with less water present (Figure 6a–d, Table 4). Because of the implemented zeolite’s high sorption capacity and its pozzolanic activity (resulting from the presence of active SiO_2_ and Al_2_O_3_ from the zeolite addition), there is a higher variety of CSH and CSAH phases [11,34].

Under the impact of the zeolite additive, the solutions become more alkaline, which leads to the hydration of the minerals from the cement clinker. Reactions (10) and (11) [12] indicate that the production of the following anions facilitates the dissolution of the zeolite tuff in liquids with an alkaline reaction:≡Si–O–Si≡ + 8OH^−^ → 2[SiO(OH)_3_]^−^ + H_2_O(10)
≡Si–O–Al≡ + 7OH^−^ → [SiO(OH)_3_]^−^ + [Al(OH)_4_]^−^(11)

Pozzolanic reactions can occur in the presence of water when the zeolite’s octahedral and tetrahedral lattice structure fragments and the cement’s calcium dioxide are present. This gives rise to the growth of hydrated calcium aluminosilicate compounds (CSAH) and facilitates the creation of natural centers of crystallization, as exemplified in part by the following potential reactions (12) and (13) [12,24,53]:

Formation of ettringite with pozzolanic reaction:6Ca^2+^ + 2[Al(OH)_4_]^–^ + 4OH^−^ + 3SO_4_^2−^ + 26H_2_O → Ca_6_Al_2_(SO_4_)_3_(OH)_12_·26H_2_O(12)

Formation of hibschite with pozzolanic reaction:Ca_3_Al_2_O_6_ + [SiO(OH)_3_]^–^ + OH^−^ + SiO_2_ → Ca_3_Al_2_(SiO_4_)_3−x_(OH)_4x_ + O^2−^, where x = 1(13)

The PXRD data and the DRIFTS identification of the newly formed crystal hydrate phases imply that there is a deficiency of bound water in their structure. The starting water-to-cement ratio and the zeolite additive’s inclusion in the second series of cement mortar compositions serve as the foundation for this assertion. As noted by other writers, zeolite’s distinct sorption characteristics aid in the development of a strong and dense structure [12,24].

The results obtained are also validated by scanning electron microscopy studies of the cement mortar morphology from the two series of samples.

## 6. Conclusions

In this study, an investigation was conducted on the changes in the microstructure of hardened mortars due to the replacement of 10 wt% of pure white cement with thermoactivated natural zeolite with high clinoptilolite content. Experiments carried out with plain sand and water–cement ratios complying with EN 197-1 show that the inclusion of zeolite increases the amount of pores accessible by mercury intrusion porosimetry by about 40%, but the measured strengths are also higher by more than 13%. When these samples were aged in an aqueous environment from day 28 to day 120, the amount of pores decreased by about 10% and the corresponding compressive strength increased by nearly 15%, i.e., the addition of thermally activated zeolite did not block the access of water to the cementitious minerals. Through microstructural studies, it was proven that new products such as ettringite, monosulphoaluminate, CSH gel—hillebrandite and tobermorite, and CSAH gel—yugawaralite, hibschite and scolecite are formed under delayed hydration. Replacing part of the white cement with thermally activated natural zeolite increases the density and strength of the structure as well as the amount and variety of new crystalline phases. This is a reason to claim that building solutions with the used zeolite content will have improved corrosion resistance against aggressive weathering.

## Figures and Tables

**Figure 1 materials-17-04798-f001:**
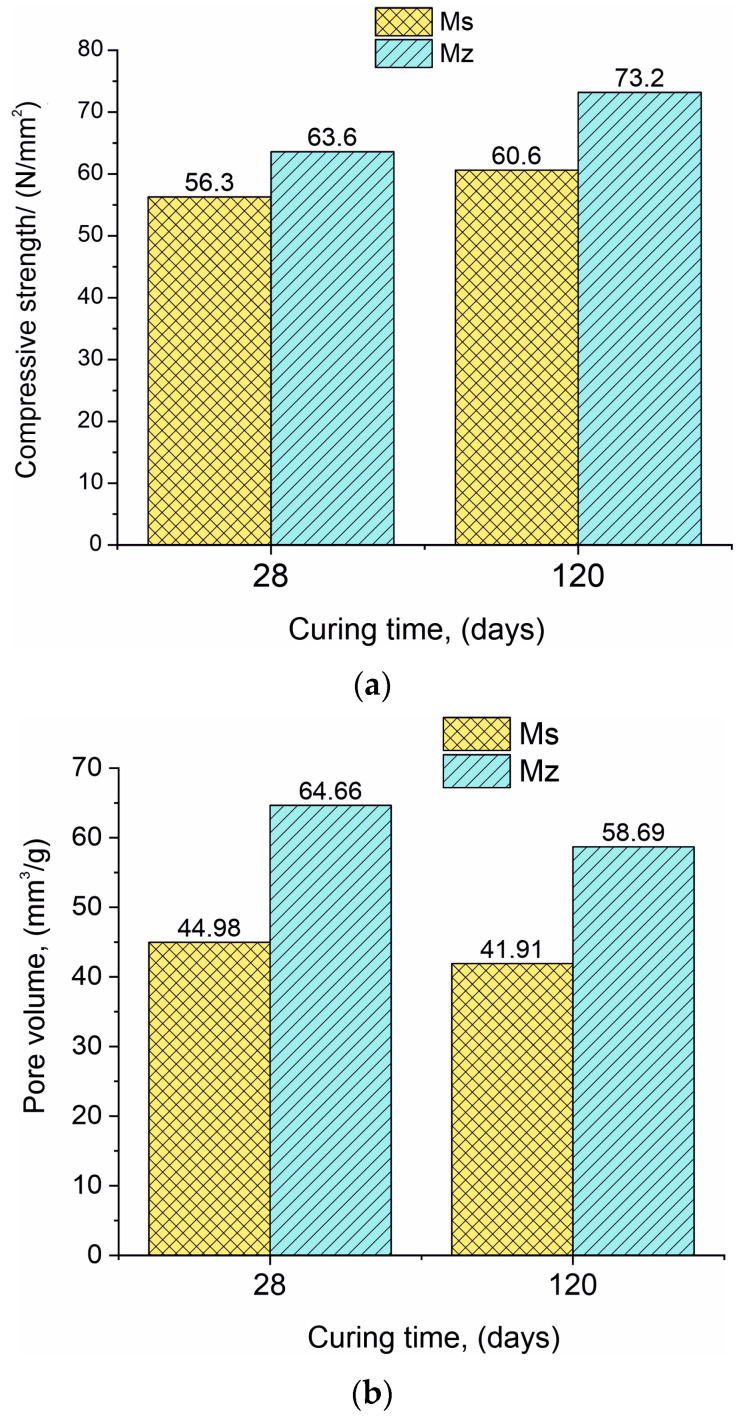
(**a**). Data presenting the compressive strength for samples Ms and Mz. (**b**). Data presenting the pore volume for samples Ms and Mz.

**Figure 2 materials-17-04798-f002:**
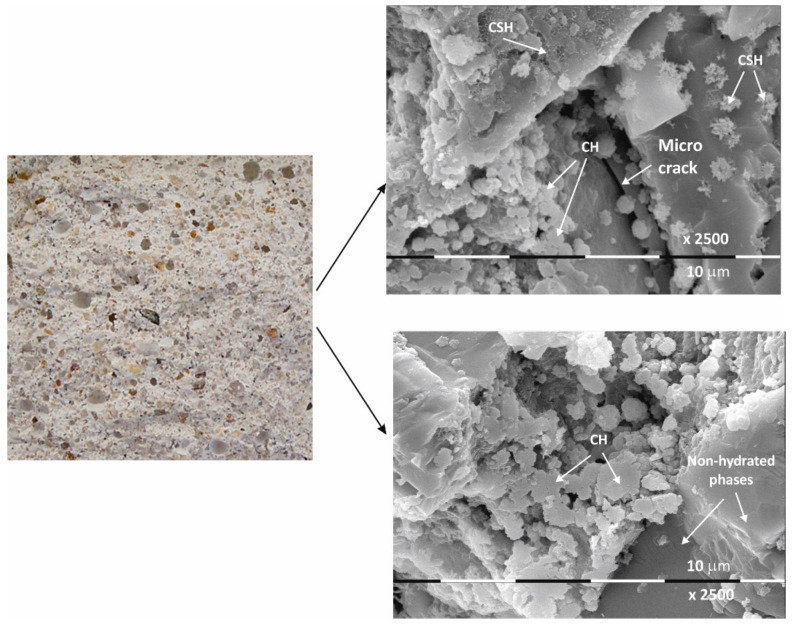
Macro- and micrographs of the surface structures of sample Ms at 28 days of water curing.

**Figure 3 materials-17-04798-f003:**
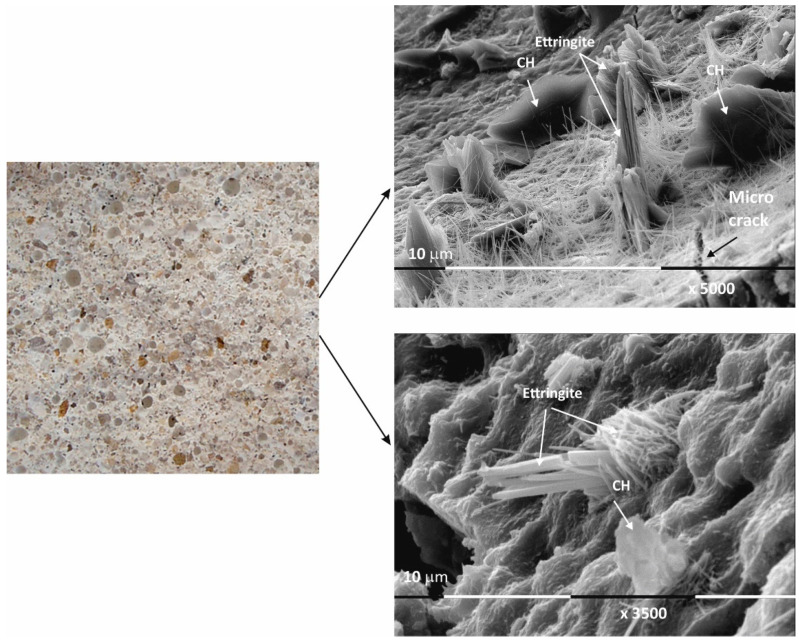
Macro- and micrographs of the surface structures of sample Ms at 120 days of water curing.

**Figure 4 materials-17-04798-f004:**
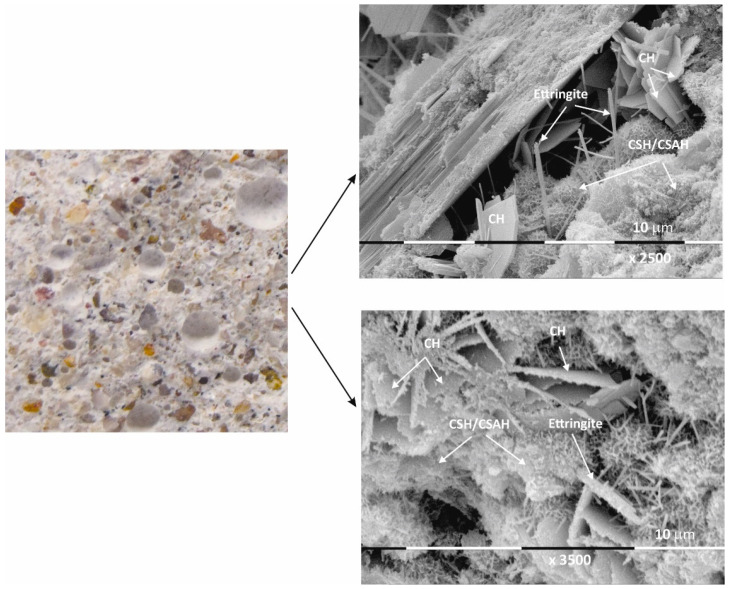
Macro- and micrographs of the surface structures of sample Mz at 28 days of water curing.

**Figure 5 materials-17-04798-f005:**
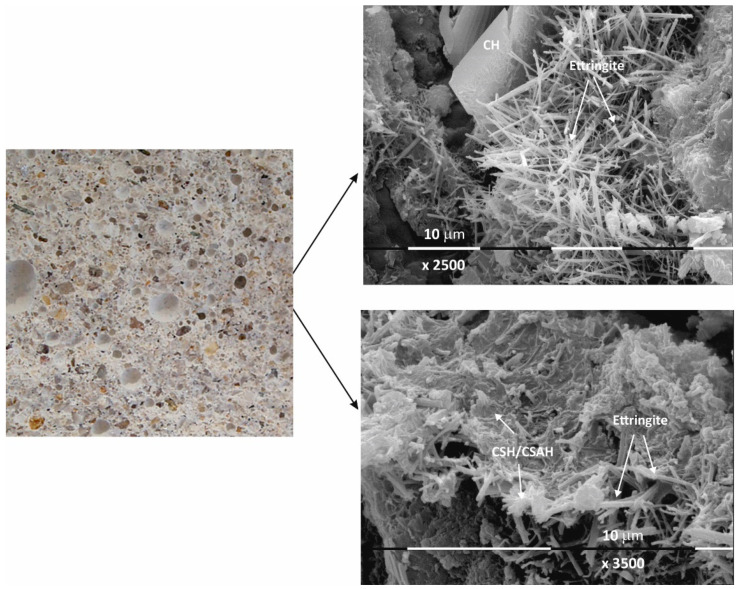
Macro- and micrographs of the surface structures of sample Mz at 120 days of water curing.

**Figure 6 materials-17-04798-f006:**
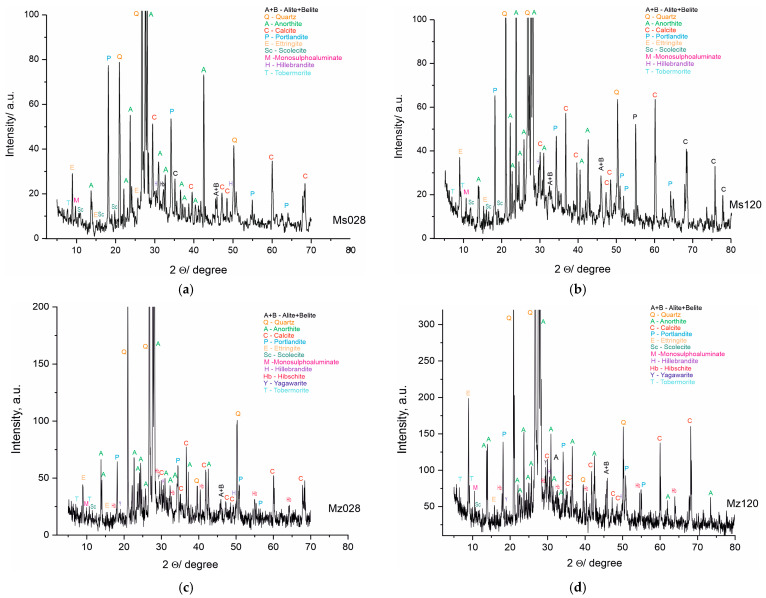
(**a**). PXRD pattern of sample Ms, water-cured for 28 days. (**b**). PXRD pattern of sample Ms, water-cured for 120 days. (**c**). PXRD pattern of sample Mz, water-cured for 28 days. (**d**). PXRD pattern of sample Mz, water-cured for 120 days.

**Figure 7 materials-17-04798-f007:**
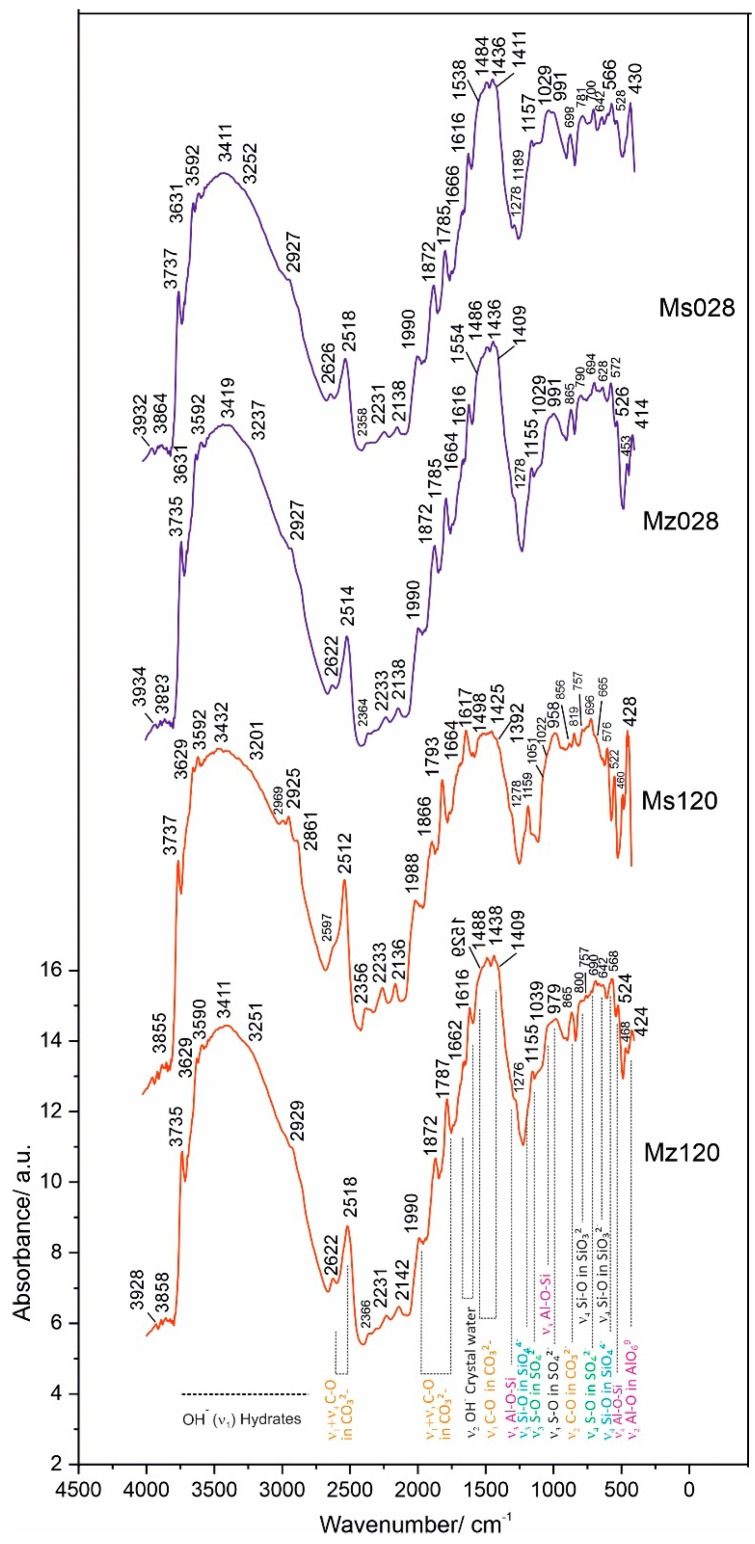
DRIFTS spectra of samples Ms28, Ms120, Mz28, and Mz120.

**Figure 8 materials-17-04798-f008:**
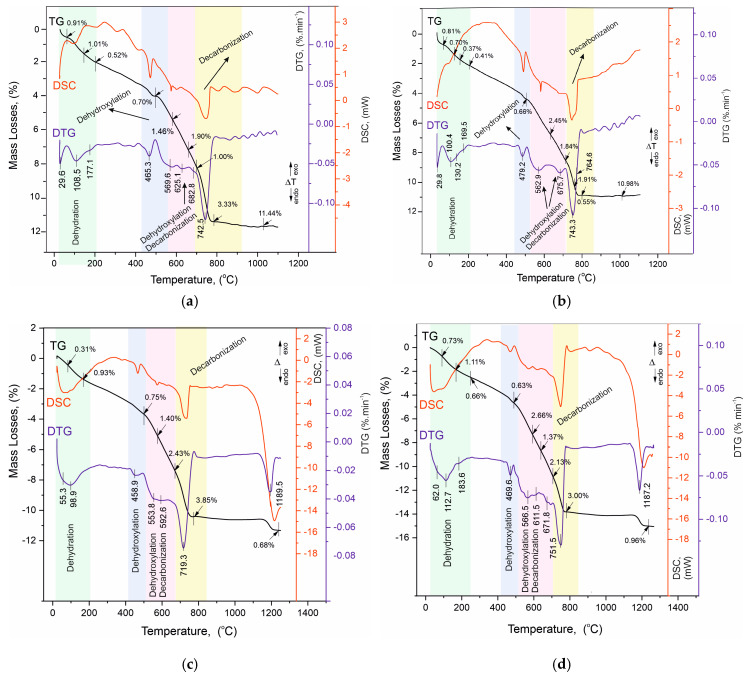
(**a**). TG/DTG-DSC curves of sample Ms28. (**b**). TG/DTG-DSC curves of sample Ms120. (**c**). TG/DTG-DSC curves of sample Mz28. (**d**). TG/DTG-DSC curves of sample Mz120.

**Figure 9 materials-17-04798-f009:**
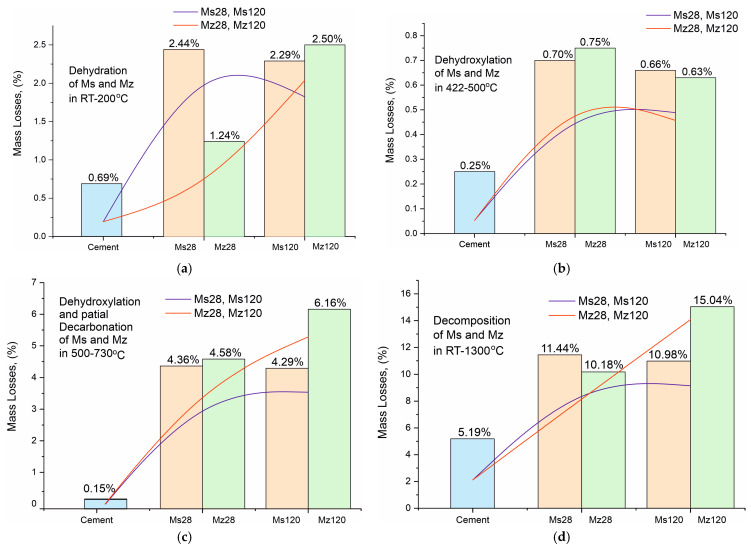
(**a**). Mass loss dynamics during dehydration of crystallization water in the RT–200 °C temperature range. (**b**). Mass loss dynamics during dehydroxylation of structurally bound water by Ca(OH)_2_ in the temperature range 422–500 °C. (**c**). Mass loss dynamics during dehydroxylation and partial decarbonation of CSH/CSAH in the temperature range 500–730 °C. (**d**). Total mass loss dynamics in the temperature range RT–1300 °C.

**Table 1 materials-17-04798-t001:** Current prices of cement substitutes.

No	Type of Cement Replacement Material, Specification	Available	Price/t	Reference
1.	Clinoptilolite, 85% purity	Local	129 €	1 *
2.	Metakaolin	Local	3200 €	2 *
3.	Silica fume	Import	300 $	3 *
4.	Pumice	Local	7500 €	4 *

1 *—http://dprao.bg/images/%D0%94%D0%BE%D0%B3%D0%BE%D0%B2%D0%BE%D1%801.pdf (in Bulgarian) (accessed on 22 September 2024). 2 *—https://nmarket.bg/metakoalin (accessed on 22 September 2024). 3 *—https://www.globalsources.com/searchList/products?keyWord=silica+fume&pageNum=1 (accessed on 22 September 2024). 4 *—https://markita.net/perlit.html-0 (accessed on 22 September 2024).

**Table 2 materials-17-04798-t002:** Abbreviations and composition of mortars.

Mortar	Cement	Zeolite	Binder	Sand	Ratios (wt–wt)		
	wt Parts	wt Parts	wt Parts	wt Parts	Binder/Sand	Water–Binder	Water–Cement
Ms Ms28 * and Ms120	1.0	-	1.0	3.0	1:3	0.50	0.50
Mz Mz28 and Mz120	0.9	0.1	1.0	3.0	1:3	0.50	0.45

* Abbreviation meaning: M—mortar; s—sand; z—zeolite. The number in the notations Ms28, Ms120, Mz28 and Mz120 indicates the duration of sample curing in days.

**Table 3 materials-17-04798-t003:** XRF analysis data of the main components of the initial raw materials and water-cured samples—Ms28, Ms120, Mz28, and Mz120.

Content of Components	White Cement	River Sand	Natural Zeolite	Sample Ms28	Sample Ms120	Sample Mz28	Sample Mz120
-	%	%	%	%	%	%	%
CaO	72.997	4.377	4.361	29.264	27.772	27.683	30.848
SiO_2_	16.857	71.545	74.727	52.442	54.234	53.580	50.730
Al_2_O_3_	3.452	12.875	13.031	9.277	9.642	9.589	9.344
MgO	0.868	1.048	0.795	1.055	0.584	0.794	1.034
Na_2_O	0.351	2.446	0.767	1.590	1.514	1.588	1.481
K_2_O	0.671	5.073	4.881	3.343	3.419	3.783	3.632
Fe_2_O_3_	0.220	1.948	1.034	0.965	0.880	0.998	0.917
SO_3_	4.312	0.100	0.023	1.560	1.581	1.539	1.569
P_2_O_5_	0.112	0.141	0.038	0.112	0.098	0.103	0.101
TiO_2_	0.091	0.282	0.141	0.231	0.147	0.206	0.191

**Table 4 materials-17-04798-t004:** Physical-mechanical properties of the samples.

Sample	Bulk Density after Immersion	Bending Strength	Adsorption after Immersion
	kg/m^3^	N/mm^2^ (MPa)	mm^3^/cm^3^
Ms	2126	7.84	173.8
Mz	2139	7.80	162.4

These values were determined according EN 1015-3:2001. Methods of test for mortar for masonry—Part 3: Determination of consistence of fresh mortar (by flow table).

**Table 5 materials-17-04798-t005:** PXRD analysis results.

No	Description	Sample	Identified Phases
1.	Non-hydrated cement phases	Ms28, Ms120, Mz28, Mz120	Belite (C_2_S), 49-1673—2CaO·SiO_2_
			Albite (C_3_S), 11-0593—(Na,Ca)Al(Si,Al)_3_O_8_
			Anorthite (CAS_2_), 41-1486—CaO·Al_2_O_3_·2SiO_2_
			Quartz, 46-1045—SiO_2_—3.34
2.	Initial component phases	Ms28, Ms120, Mz28, Mz120	Mg-rich Calcite, 47-1743—CaCO_3_
3.	Newly formed phases: containing OH^−^	Ms28, Ms120, Mz28, Mz120	Portlandite (CH), 44-1481—Ca(OH)_2_
3.1.	containing SO_4_^2−^, OH^−^, and crystal water H_2_O	Ms28, Ms120, Mz28, Mz120	Ettringite, 41-1451—Ca_6_Al_2_(SO_4_)_3_(OH)_12_·26H_2_O
		Ms28, Ms120, Mz28, Mz120	Monosulphoaluminate, 45-0158—Ca_4_Al_2_SO_10_·12H_2_O
3.2.	hydroxyl silicates—CSH, formed from the main oxides CaO, SiO_2_, OH^−^, and/or crystal water H_2_O	Ms28, Ms120, Mz28, Mz120	Hillebrandite, 42-0538—Ca_6_Si_3_O_9_(OH)_6_
		Ms28, Ms120, Mz28, Mz120	Tobermorite 11A, 45-1480—Ca_5_Si_6_(O,OH)_18_·5H_2_O
3.3	hydroxyl silicates—CSAH, formed from the main oxides CaO, Al_2_O_3_, SiO_2_, OH^−^, and/or crystal water H_2_O	Mz28, Mz120	Yugawaralite, 39-1372—CaAl_2_Si_6_O_16_·4H_2_O
		Mz28, Mz120	Hibschite, 45-1447—Ca_3_Al_2_(SiO_4_)_3−x_(OH)_4x_; (x = 0.2–1.5)
		Ms28, Ms120, Mz28, Mz120	Scolecite, 41-1355—CaAl_2_Si_3_O_10_·3H_2_O

**Table 6 materials-17-04798-t006:** DRIFTS spectroscopy results of samples Ms and Mz.

No	Description/ References	Bond	ν_1_ (cm^−1^)	ν_2_ (cm^−1^)	ν_3_ (cm^−1^)	ν_4_ (cm^−1^)	Samples
1.	AlO_6_^9−^ [44,45,46]	Al-O	-	428–430	-	-	Ms
			-	414–424	856	-	Mz
2.	SiO_3_^2−^ [44,45,46]	Si-O	-	468–472		781	Ms28
			-	470–472	1041–1053	790	Mz28
3.	SiO_4_^4−^ [44,45,46]	Si-O	-	-	1189	566–576	Ms
					-	572–576	Mz
4.	SO_4_^2−^ [44,45,46]	S-O	958–991	-	1155–1159	696–700 781	Ms
			979–991	-	1155	690–694 790–800	Mz
5.	CO_3_^2−^ [45,46,47]	C=O	-	869	1411–1425 1425–1436 1484–1498 1538	-	Ms
			-	865	1409 1436–1438 1486–1488 1529–1554	-	Mz
			ν_1_ + ν_3_ 2524 2364–2622	-	-	ν_1_ + ν_4_ 1785–1793 1866–1872 1988–1990	Ms
			ν_1_ + ν_4_ 2524–2526 2588–2628	-	-	ν_1_ + ν_4_ 1785–1787 1873–1990	Mz
6.	SiO_4_^4+^-AlO_6_^9−^ [48]	Si-O-Al	-	798–800	1022–1029 1278	522–528	Ms
			-	-	1029–1039 1276–1278	524–526	Mz
7.	SiO_2_	Si-O	-	-	796–804	-	Ms
8.	O-H struct. ^(1)^ [49]	O-H	-	-	3592	-	Ms
			-	-	3590–3592	-	Mz
9.	O-H^−^ cryst. ^(2)^ [49]	O-H	3411–3432	1616–1666	3201–3252	-	Ms
			3411–3419	1616–1662	3237–3251	-	Mz
10.	O-H^−^ [44,49]	Ca-OH	-	-	3629–3631	-	Mz, Ms
11.	O-H^−^ [50]	Al-OH	-	-	3737	-	Ms
			-	-	3735	-	Mz

O-H struct. ^(1)^—O-H bond in structural OH^−^ anion [49]. O-H crystal. ^(2)^—O-H bond belonging to crystal-bonded water molecules [49].

**Table 7 materials-17-04798-t007:** Thermal investigation results based on temperature at inflexion point (T_infl._) and mass loss (ML) of the raw material, white Portland cement, and samples Ms28, Ms120, Mz28, and Mz120.

Sample	1. TR *—30–200 °C		2. TR *—422–500 °C		3. TR *—500–730 °C		4. TR *—640–800 °C		ML^Tot^ **, (%)
	T_infl._, (°C)	ML, (%)	T_infl._, °C	ML, (%)	T_infl._, (°C)	ML (%)	T_infl._ (°C)	ML (%)	
Cement [26]	65.1 98.0	0.30 0.39	429.9	0.25	539.6	0.15	711.8	3.68	5.19
Ms28	29.8 108.5 177.1	0.91 1.01 0.52	465.3	0.70	569.6 625.1 682.8	1.46 1.90 1.00	742.5	3.33	11.44
Ms120	29.8 100.4 130.2 169.5	0.81 0.70 0.37 0.41	479.2	0.66	562.9	2.45	675.7 743.3	1.84 1.91	10.98
Mz28	55.3 98.92	0.31 0.93	458.9	0.75	553.8 592.6	1.40 2.43	719.3	2.89	10.18
Mz120	62.0 112.7 183.6	0.73 1.11 0.66	462.6	0.63	566.5 611.5 671.8	2.66 1.37 2.13	751.5	3.00	15.04

TR *—temperature range, (°C); ML^Tot^**—Total mass losses, (%).

## Data Availability

The original contributions presented in the study are included in the article, further inquiries can be directed to the corresponding author.

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
