# Peer review of "A Study of the Influence of Thermoactivated Natural Zeolite on the Hydration of White Cement Mortars"

_materials, 2024, doi:10.3390/ma17194798_

Round 1

Reviewer 1 Report

Comments and Suggestions for Authors

See file.

Author Response

No new comments from the reviewer after R1.

Reviewer 2 Report

Comments and Suggestions for Authors

The article presents a study on natural pozzolan on the hydration of white cement mortars. Based on the information in the manuscript, it is possible to identify the potential for publication of the article, as the topic is very prominent in the area of ​​materials. However, some points need to be clarified before the article is accepted:

- Abstract: it would be important to include numerical results in this section. Additionally, include the main conclusions obtained in your research.

- In the text of the manuscript it is not clear why the authors chose to carry out the research with White cement. I think that based on the information highlighted in your text, the methodology would be applicable if conventional cement were used. Why choose White cement? Please justify this information in the text of the manuscript.

- In the introduction, the authors use a lot of information summarized by (); I think the writing of the article would be richer if the authors explained the information, not just citing the main points through parentheses. Please check the language pattern used in the search.

- “Among the available pozzolans (tuffs, ashes, perlites, pumices, zeolites), zeolite in particular is considered to be an excellent supplementary cementitious material [9].” In this information, I suggest that the authors substitute natural pozzolan for natural and include a clear mention of volcanic ash, the first natural pozzolan studied and applied on a large scale.

- “The standardized method to evaluate the pozzolanic activity is the Frattini test, which involves chemical titration to determine the dissolved Ca2+ and OH− concentrations in a solution containing cement and test pozzolan [19,20].” This method has advantages and disadvantages, like all methods used to evaluate pozzolanicity of materials. I miss the authors commenting in more depth on the main methods used in pozzolanicity analysis, see for example the information present in Methods for Evaluating Pozzolanic Reactivity in Calcined Clays: A Review and adequately discussing the advantages and disadvantages of the method described in its introduction.

- “Moreover, the durability of zeolite containing cement mixtures is improved due to the presence of a finer pore matrix with decreased amount of C3A and CaO containing hydration products with lower leachability and reduction in SO3 binding into the cement paste [17,18,22 ].” I agree with your point. However, I miss the authors highlighting the pozzolanic reaction and what this reaction represents in terms of durability. It is known that the reactive amorphous silica present in pozzolanic materials reacts with Ca(OH)2 – portlandite and produces greater formation of C-S-H. This is an important point from the point of view of durability, since portlandite is very reactive and responsible for mechanisms of degradation of the cement matrix, such as carbonation, leaching, chloride attack, for example. This has been discussed in some experimental articles, see for example the introduction to Silica fume activated by NaOH and KOH in cement mortars: Rheological and mechanical study. I think it would be important to include this type of discussion in the text of the manuscript, because it highlights the importance of your study. Please comment about this in your review.

- “Natural pozzolans are an attractive alternative, since they can be found in large volumes, especially in certain geographical locations [10].” This is an important point of view, but note that when non-natural pozzolans are used from residues from other production cycles (such as fly ash or ash from agro-industrial residues), another important advantage is included in the production of cement or concrete: substitution If the binder is Portland cement, whose production is highly polluting, for waste that would have no applications, therefore the environmental gain is twofold. In the case of natural pozzolans this does not happen. I miss the authors commenting on this point in the text of the manuscript. Please consider including this in your article discussion.

- Table 1: why did the authors choose to evaluate the 1:3:0.50 composition? Please justify this information using technical criteria, normative procedures or references to works with similar compositions. Justify your choice in this composition.

- Table 2: compare the results of the chemical composition of Natural Zeolite with other pozzolanic materials.

- Table 3: I think some of this information would be better presented in the form of figures rather than tables. Please consider this in the text of the manuscript. Furthermore, use the units commonly used in international bibliography, for example, change from N/mm² to MPa.

- Figure 1: I have a doubt regarding the results of this figure. Hydration reactions are packing reactions, that is, over time the mortars become more compact, unless there is some degradation mechanism, because the formation of C-S-H is increased, improving the packing of the material. However, observing the results of your research, it is observed that the pore volume increases from 28 to 120 days, both in the Ms and Mz composition. Why does this happen? Please justify your information. Also note that there is an error in figure 1, change it to Ms, 28 days and Mz, 28 days; and Ms, 120 days and Mz, 120 days. Please correct this information.

- Figures 2-5: these results are so interesting and useful in your research. However, I expected that the compositions containing Mz would present smaller amounts of portlandite, why did this not happen?

- “In conclusion, due to the different cement-to-water ratio and empty space, these area are filled with crystal hydrates of different morphology: plate crystals (portlandite), needle crystals (ettringite) and fine crystal aggregates (CSH/CSAH gels ).” This information is very useful, please correlate this information with the Ms and Mz compositions.

- Figure 6: according to the methodology, the scan would be from 5 to 80º. Some results appear to have a different scan range. Please check.

- Discussion: the authors highlight some reactions with “only in compositions with sand or zeolite”. This doesn't seem appropriate because it appears that sand and zeolite have the same effects, but note that sand has no chemical effect, only a physical one. Review this information.

- The conclusions are bad, use the information in topics and highlight the most important points of your research. If necessary, change as per previous comments.

Author Response

The reviewer has accepted all our R1 responses as satisfactory and has no further comments to make.

Reviewer 3 Report

Comments and Suggestions for Authors

The paper by Stoyanov and coworkers presents a study about the addition of natural zeolites to cements. It is an interesting study in which the phase formation has been studied by XRF, XRD, DRIFTS and SEM, that deserves publication.

I would suggest including a brief economic study about including natural zeolites. May be just a paragraph with a table with the prizes of natural zelolites versus other conventional raw materials used for the fabrication of cements. 

Please revise the significative numbers used in Table 2 and keep in mind the measurement error of the XRF equipment.

Since the DRIFTS cell can be heated and it is closed (so synthetic air or any other gas can be feed), please indicate the conditions in which the DRIFTS spectra have been recorded. 

Author Response

(The authors gave the same response as above.)

Round 2

Reviewer 2 Report

Comments and Suggestions for Authors

Accept in present form

Author Response

No new comments from the reviewer after R1